# Vertical aerosol particle exchange in the marine boundary layer estimated from helicopter-borne measurements in the Azores region

**Janine Lückerath**[1,a]**, Andreas Held**[2,a]**, Holger Siebert**[1]**, Michel Michalkow**[1]**, and Birgit Wehner**[1]

[1]Leibniz Institute for Tropospheric Research, Leipzig, Germany
[2]Environmental Chemistry and Air Research, Technische Universität Berlin, Berlin, Germany
[a]previously at: Atmospheric Chemistry, University of Bayreuth, Bayreuth, Germany

**Correspondence:** Birgit Wehner (birgit@tropos.de)

**Abstract.** Aerosol particles are important for radiation effects, cloud formation, and therefore the climate system. A detailed understanding of the spatial distribution of aerosol particles within the atmospheric boundary layer, which depends on sources and sinks, as well as long-range transport and vertical exchange, is important. Especially in marine regions, where the climate effect of clouds is comparably high, long-range transport with subsequent vertical mixing dominates over local aerosol sources.

In this study, three different methods were applied to estimate the vertical aerosol particle flux in the marine boundary layer (MBL) and the vertical exchange between the MBL and the free troposphere (FT): eddy covariance (EC), flux–gradient similarity ($K$ theory), and the mixed-layer gradient method (MLG). For the first time, MBL aerosol fluxes derived from these three methods were compared in the framework of the "Azores Stratocumulus Measurements of Radiation, Turbulence and Aerosols" (ACORES) field campaign in the Azores region in the northeastern Atlantic Ocean in July 2017. Meteorological parameters and aerosol and cloud properties were measured in the marine troposphere using the helicopter-borne measurement platform ACTOS (Airborne Cloud Turbulence Observation System).

All three methods were applied to estimate the net particle exchange between MBL and FT. In many cases, the entrainment fluxes of the MLG method agreed within the range of uncertainty with the EC and $K$-theory flux estimates close to the top of the MBL, while the surface flux estimates of the different methods diverged. It was not possible to measure directly above the surface with the helicopter-borne payload, which might be a source of uncertainty in the surface fluxes. The observed particle fluxes at the top of the MBL ranged from 0 to $10 \times 10^6 \, \mathrm{m^{-2} \, s^{-1}}$ both in the upward and the downward direction, and the associated uncertainties were on the same order of magnitude. Even though the uncertainties of all three methods are considerable, the results of this study contribute to an improved understanding of the transport of particles between the MBL and FT and their distribution in the MBL.

## 1 Introduction

Aerosol particles influence the global climate in different ways: (i) they scatter and absorb the incoming solar radiation, and (ii) they may act as cloud condensation nuclei (CCN) and affect cloud microphysical properties. These effects have been studied intensively over the last few decades; however, there is still a particularly large uncertainty relating to the influence of the number size distribution on cloud droplet number distribution (Stevens and Feingold, 2009) and the associated change in the shortwave albedo of clouds (Twomey, 1977; Ackerman et al., 2000; Werner et al., 2014).

Meanwhile, a comprehensive network of measurement stations for aerosol monitoring using in situ and remote sens-

ing methods in industrialized, continental regions has been established (e.g., ACTRIS, http://www.actris.net/, last access: 11 July 2022), but marine areas are still poorly characterized. However, more than 70 % of the Earth's surface is covered with water, and these regions have a significant impact on the global climate. Furthermore, climate models indicate that a large fraction of the aerosol indirect radiative forcing is associated with marine low clouds (Quaas et al., 2009), while the simulation of these clouds is still very uncertain in climate models (Wyant et al., 2010). For the remote marine atmosphere, local anthropogenic emissions play a minor role. While the larger accumulation mode (diameter > 300 nm) is dominated by sea spray aerosol, the contribution of sea spray to the particle diameter range smaller than 300 nm is evaluated differently in the literature (Zheng et al., 2018; Ovadnevaite et al., 2014; de Leeuw et al., 2011) and is therefore subject of further research. The major fraction of aerosol particles in the marine boundary layer (MBL) originates in continental regions and is transported mainly in the free troposphere (FT) over long distances (Clarke et al., 2013; Logan et al., 2014) or is formed in the FT via new particle formation, e.g., in the outflow of deep convective clouds (e.g., Clarke et al., 2013). To become active as CCN in the MBL they need to be mixed downwards, i.e., they have to pass the inversion layer. The vertical mixing between these layers is the crucial process and needs to be quantified to understand the vertical distribution of aerosol particles and the evolution of them in the marine boundary layer. This has been done for tropical regions, e.g., by Clarke et al. (1996), resulting in entrainment rates of $0.6\,\mathrm{cm\,s^{-1}}$ into the MBL. This leads to an effective transport of potential CCN from FT to MBL (Clarke et al., 2013). However, these studies were performed in the Intertropical Convergence Zone where vertical exchange is very strong. For other regions, such as the midlatitudes, experimental studies investigating the vertical mixing of aerosol between FT and MBL have been lacking. Katoshevski et al. (1999) used a box model to evaluate the influence of sources and sinks on the aerosol budget in remote marine regions and concluded that nucleation and further particle growth play a crucial role. The exchange between FT and MBL affects the aerosol dynamics in the subtropical MBL and thus also CCN concentrations (Raes, 1995). The influence of aerosol particles on the dynamics and structure of marine stratocumulus clouds remains poorly understood and needs to be studied in more detail (Wood, 2012). This includes both the long-range transport and the vertical mixing into the MBL.

Aerosol in situ measurements in the MBL are limited to ship- or aircraft-based short-term campaigns and/or those performed on islands in the ocean. Previous studies were performed, for example, near Tasmania (e.g., Bates et al., 1998; Clarke et al., 1998), between the Canary Islands and Portugal (e.g., Raes et al., 2000), in the northeastern Atlantic Ocean (e.g., Norris et al., 2012; Petelski and Piskozub, 2006), on Christmas Island in the equatorial Pacific (e.g., Clarke et al., 2013), and over the Azores (e.g., Dong et al., 2014, 2015; Wood, 2012; Wood et al., 2015). Results from sea spray emission studies are published in, e.g., Geever et al. (2005); de Leeuw et al. (2011) and Ovadnevaite et al. (2014). Particle number fluxes during nucleation events at the Irish Atlantic coast were studied by Flanagan et al. (2005), while Ceburnis et al. (2016) investigated sources and sinks of aerosol particles at the same location. The Azores are the only site located between the subtropics and the midlatitudes in the northern Atlantic Ocean that is representative for a large fraction of marine areas.

In previous studies it turned out that the islands of Azores provide a good location for studying the MBL with low anthropogenic influence. For this purpose, the permanent ENA ARM (Eastern North Atlantic Atmospheric Radiation Measurement) site has been established on the island of Graciosa (e.g., Dong et al., 2014; Wood et al., 2015). For a better understanding of vertical transport processes in the cloudy MBL and in the cloud-free MBL the "Azores Stratocumulus Measurements of Radiation, Turbulence and Aerosols" (ACORES) project was initiated, and the intensive campaign was performed in July 2017.

One common method to estimate the vertical transport of aerosol particles is eddy covariance (EC) combined with condensation particle counters, which has been applied in earlier studies using fixed-point measurements (e.g., Buzorius et al., 1998). Buzorius et al. (2006) published a first pilot study estimating a vertical particle flux via the eddy covariance method, using an aircraft as the measurement platform. One advantage of an airborne platform is the possibility of making measurements in different levels of the boundary layer or close to inversion layers. One challenge for estimating fluxes is to fulfill the criteria for stationarity and homogeneity, which is often not fulfilled for horizontal flight patterns. To our knowledge, until now only Buzorius et al. (2006) have used aircraft measurements to calculate particle fluxes via the EC method. Due to the relatively high flight speed and limited time resolution of measurements, the uncertainties were quite high.

In our study, we use a slow-flying helicopter in combination with highly resolved measurements under conditions with low anthropogenic influence. In addition to the EC method, two gradient-based methods are applied to calculate vertical turbulent particle fluxes from profile measurements in the MBL above the northeastern Atlantic Ocean.

## 2 Methods

### 2.1 ACORES 2017 campaign

In July 2017, the "Azores Stratocumulus Measurements of Radiation, Turbulence and Aerosols" (ACORES) campaign was performed in the northeastern Atlantic Ocean at the islands of Azores. The archipelago is located approximately 1400 km west of the European continent. During the

ACORES campaign ground-based measurements of aerosol particle number concentration and size distribution were performed at the ENA (eastern North Atlantic) ARM site on the island of Graciosa at sea level and at the Pico Mountain Observatory (Observatorio da Montanha do Pico, OMP) in 2225 m above sea level. Helicopter-borne measurements of aerosol particle number concentration and meteorological parameters were performed from Graciosa airport up to 3000 m covering the marine boundary layer (MBL) and free troposphere (FT). More details about the campaign are given by Siebert et al. (2021).

Graciosa is a small island ($\approx 60\,\mathrm{km}^2$ area) situated at $39.1°$ N, $28.0°$ W in the Azores archipelago, at a latitude between the subtropics and the midlatitudes. As such, Graciosa is influenced by different meteorological conditions, including periods of relatively undisturbed trade wind flow, mid-latitude cyclonic systems and associated fronts, and periods of extensive low-level cloudiness. The ACORES campaign took place from 2 until 23 July. According to meteorological conditions the campaign was divided into three periods: (1) until 11 July, dominated by dry air with low cloud fraction; (2) 12–19 July, with warm and humid air masses and frequent precipitation; and (3) after 20 July, again with dry conditions with low cloud fraction (Siebert et al., 2021). The ACORES measurement strategy allowed for so-called aerosol flights focusing on the vertical stratification and transport of aerosol particles under conditions with no or few clouds. The airborne measurements were performed using the helicopter-borne platform ACTOS (Airborne Cloud Turbulence Observation System) (Siebert et al., 2006) as external cargo hanging 170 m below a helicopter. The advantage of the system is the low true air speed of $20\,\mathrm{m\,s^{-1}}$ leading to a higher spatial resolution compared to fast-flying aircraft. During ascent and descent, the helicopter has always a true airspeed of about $20\,\mathrm{m\,s^{-1}}$, and the measurements are not influenced by the helicopter (Siebert et al., 2006). Furthermore, technical requirements such as inlet design and sampling issues are less demanding compared to fast-flying aircraft.

Each measurement flight started with a vertical profile up to a height well above the inversion layer followed by a specific flight pattern according to the meteorological conditions. Under low-cloud or cloud-free conditions, horizontal legs were flown at constant heights. These vertical profiles and horizontal legs are the main database of this study.

## 2.2 Instruments and data

An overview over the instrumentation and specifications used on ACTOS is given in Siebert et al. (2021). Only parameters used in this study will be explained here in more detail. ACTOS is equipped with instruments to measure basic meteorological parameters with high temporal resolution, such as the 3D wind vector (ultrasonic anemometer, Solent HS Gill), absolute humidity (Dew Point Mirror, TP3, Meteo-

Labor AG), and temperature (PT100, Rosemount Series 139 plus ultrafast airborne thermometer, UFT), as well as cloud properties such as liquid water content (LWC), which is not subject of this study.

The total particle number concentration is measured with a commercial condensation particle counter (CPC Model 3762A, TSI) (TSI, 1996) with a modified lower cut-off diameter of 8.5 nm and a modified flow rate of $1\,\mathrm{L\,min^{-1}}$. CPC data have been sampled and post-processed with a time resolution of 0.1 s. However, note that the typical response time of this CPC is approximately 1 s. Aerosol number concentrations are corrected for losses in the inlet system using a mean factor for diameters between 20 and 1000 nm that has been determined experimentally and for variations in the sample flow due to pressure changes. Furthermore, all aerosol data are transferred to standard conditions of $T = 288$ K and $p = 1013.15$ hPa.

Although the wind vector measured in the ACTOS reference system has been transferred to an Earth-fixed system using an inertial navigation system, the typical pendulum motion is still visible with a more or less sharp frequency around 0.04 Hz. To minimize this effect, a spectral band-stop filter has been applied in the range between $0.03\,\mathrm{s^{-1}} < f < 0.05\,\mathrm{s^{-1}}$.

## 2.3 Flux estimation methods

Three different methods to calculate vertical turbulent exchange in the boundary layer (BL) are applied: the eddy covariance (EC) method, the $K$-theory method, and the mixed-layer gradient method (MLG). In order to highlight advantages and limitations of each method, a brief introduction and specific assumptions and requirements pertinent to the methods will be given.

### 2.3.1 Eddy covariance method

The EC method is a widely used micrometeorological method to directly measure turbulent vertical fluxes of atmospheric constituents through a horizontal, homogeneous plane (Businger, 1986; Foken et al., 2012).

It is based on the mass balance equation, which can be simplified due to the assumptions of stationarity and horizontal homogeneity. The vertical flux $F_{\mathrm{EC}}$ of a scalar can be estimated by correlation of the fluctuation of the vertical wind component $w'$ and the fluctuation of a scalar concentration $c'$, which is equal to the covariance of the vertical wind speed $w$ and the scalar concentration $c$:

$$F_{\mathrm{EC}} = \overline{w'c'} = \frac{1}{M-1} \sum_{k=0}^{M-1} [(w_k - \overline{w_k})(c_k - \overline{c_k})]. \qquad (1)$$

The overbar indicates the mean over a certain averaging period, which is typically 30 min for atmospheric turbulent fluxes depending on the dominating scales at a fixed location

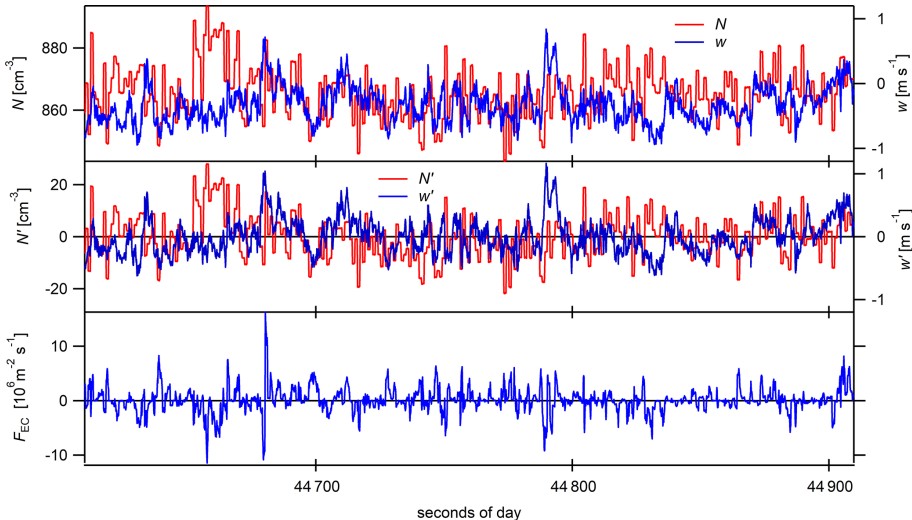

**Figure 1.** Selected data, measured during a horizontal flight leg (altitude 275 m) on 10 July 2017. The upper plot shows the time series of total particle number concentration $N$ and vertical wind speed $w$. The middle plot shows the resulting fluctuations $N'$ and $w'$. The lower plot shows the resulting time series of $w' \cdot N'$.

(e.g., tower measurements). $M$ is the number of data points in each averaging period, and $k$ indicates the data point at time $t_k$ (Foken, 2016). The scalar $c$ and the vertical wind speed have to be measured with high measurement frequency (typi-
5 cally at least 10 Hz for tower measurements) in order to cover high frequencies of the turbulent spectrum and their contribution to the flux.

Even though the EC method was developed for ground-based measurements it has also been applied to airborne mea-
10 surements before (e.g., Buzorius et al., 2006; Metzger et al., 2012; Yuan et al., 2015). Airborne sampling provides spatial averaging directly (Buzorius et al., 2006) if frozen turbulence (Taylor, 1938) is assumed, i.e., if the statistical properties of turbulence do not change. Thus, measuring at a fixed point
at the ground, e.g., for 30 min at a wind speed of $2 \, \mathrm{m \, s^{-1}}$, probes the same air mass and eddies as a 3 min flight leg at a true air speed of $20 \, \mathrm{m \, s^{-1}}$.

Figure 1 illustrates the calculation of the turbulent flux of aerosol particles from airborne measurements in this study.
The upper panel shows the time series of vertical wind speed $w$ and particle number concentration $N$ measured during a horizontal flight leg within the MBL with 1 s time resolution. In the middle panel, the fluctuation of both variables is shown according to $w' = w - \overline{w}$ and $N' = N - \overline{N}$. For the time series
of $F_{\mathrm{EC}}$ (lower panel) Eq. (1) was applied. From these time series, the mean values and standard deviations used in the following plots and tables were calculated.

Uncertainty ranges of EC particle fluxes based on counting statistics were calculated following Buzorius et al. (2003). In
order to estimate whether the EC flux estimates are significantly different from zero, the random shuffle method by Billesbach (2011) was used. This method estimates the con-

tribution of random instrument noise to the total uncertainty of the flux calculation.

### 2.3.2  *K*-theory method    35

Vertical fluxes can also be estimated using the so-called gradient approach or $K$ theory assuming stationarity and horizontal homogeneity within the BL. In this $K$ theory, closure is accomplished when the flux is linearly proportional to the mean gradient (flux–gradient similarity), and the proportion-  40
ality constant $K$ describes all properties influencing the vertical turbulent exchange:

$$F_{\mathrm{K}} = -K \frac{\mathrm{d}\overline{c}}{\mathrm{d}z}, \tag{2}$$

where $F_K$ is the vertical flux of a scalar $c$ and $\frac{\mathrm{d}\overline{c}}{\mathrm{d}z}$ is the mean gradient of the scalar with height $z$. The vertical turbulent  45
diffusivity $K$ describes the efficiency of the vertical mixing. In this study, $K$ is estimated following Hanna (1968):

$$K = 0.3 \, \sigma_{\mathrm{w}} \, l, \tag{3}$$

where $\sigma_{\mathrm{w}}$ is the standard deviation of the vertical wind. The typical length scale $l$ for the dominant eddies is defined as  50

$$l = v_{\mathrm{TAS}} \, \tau, \tag{4}$$

where $v_{\mathrm{TAS}}$ is the true air speed and $\tau$ is the time lag when the auto-correlation function of the vertical wind drops to $1/e$. In order to calculate $\sigma_{\mathrm{w}}$ and $\tau$, a horizontal flight leg within the MBL is needed to characterize turbulence. Due to the fact  55
that not all flights included horizontal flight legs, averages and standard deviations were calculated from all five available flight legs, $\sigma_w = 0.3 \pm 0.16 \, \mathrm{m \, s^{-1}}$ and $\tau = 4.8 \pm 1.7 \, \mathrm{s}$.

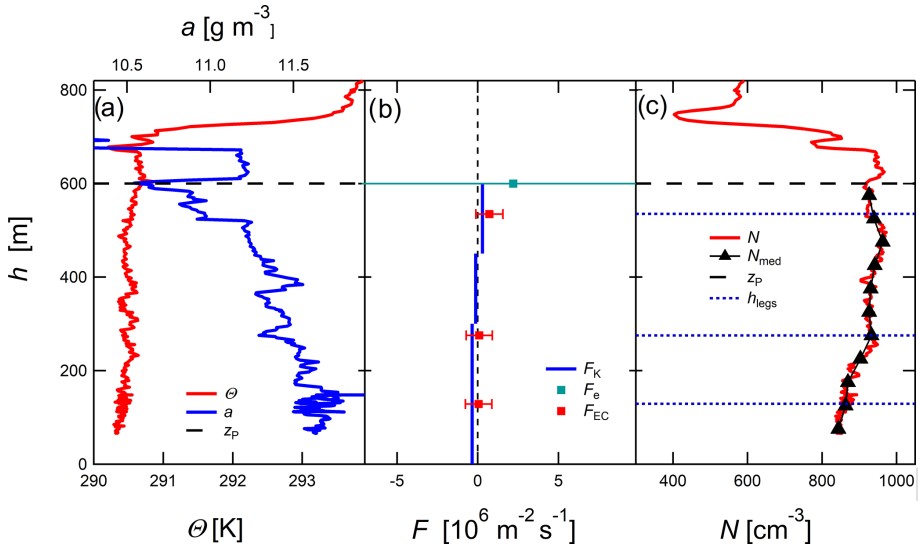

**Figure 2.** Data for flight no. 7, 10 July. **(a)** Vertical profiles of potential temperature (red) and absolute humidity (blue). The horizontal dashed black line shows the particle mixing height $z_P$. **(b)** Profiles of aerosol particle flux estimates of all three methods including uncertainties of $F_e$ and $F_{EC}$. The particle flux results of the MLG ($F_e$) and $K$ ($F_K$) methods, which are based on sections of the median profiles **(c)**, are shown in blue and green. The estimates of the EC method ($F_{EC}$) are shown in red; they are based on horizontal flight legs shown in **c**). **(c)** Profile ($N$) and median profile ($N_{med}$) of aerosol particle number concentration of the CPC (red and black) and the height of the horizontal flight legs (dotted blue lines, approximately 130, 275, and 535 m height).

These average values of $\sigma_w$ and $\tau$ were used for each flight. We note that the standard deviations of individual horizontal flight legs are approximately 53 % and 34 % of the average values, respectively.

5 In order to estimate the uncertainty of the fluxes calculated by $K$ theory, Monte Carlo simulation (MCS) (Anderson, 1976) was applied. For this purpose, the original calculation according to Eq. (2) is repeated 100 000 times with slightly changed input values in order to take into account 10 their uncertainty in a random fashion, and then the resulting flux estimates are statistically evaluated. For $K$-theory particle fluxes in this study, a 10 % uncertainty was assumed for aerosol particle number concentration ($N \pm 0.1\,N$). For $\sigma_w$ and $\tau$, the ranges given above, which correspond to the stan- 15 dard deviation of the five available values, were used. For each of these parameters, a value was taken randomly from a uniform distribution between the minimum and maximum values, and combined in one simulation of the flux calculation. This procedure was repeated 100 000 times, which led 20 to a normal distribution of 100 000 simulations with a mean value and standard deviation of the flux estimate.

It should be noted that the vertical turbulent diffusivity according to Eq. (3) assumes similar $K$ values of momentum and particle fluxes, which is a reasonable assumption 25 (Siebert et al., 2004). In $K$ theory, it is assumed that there is one mean gradient across the layer of interest. In order to determine gradients, a linear model was fitted to median profiles of particle number concentrations above the ocean within the MBL. It was applied for the whole MBL or for

linear segments of the profile in cases were obvious gradi- 30 ent changes occurred. In these cases, the estimated fluxes are representative for the selected height ranges.

### 2.3.3 Mixed-layer gradient method

The mixed-layer gradient (MLG) method is also based on flux–gradient similarity and derived from $K$ theory. It relates 35 vertical gradients of scalars $\frac{\partial c}{\partial z}$ to two fluxes, a surface flux $F_s$ and an entrainment flux $F_e$ (Lenschow et al., 1999; Wyngaard and Brost, 1984). Thus, MLG takes into account sources and sinks at two interfaces, the interface between the surface and the MBL and the interface between the MBL and the FT. 40 Based on mixed-layer scaling the turbulent equation of motion can be closed:

$$\frac{\partial c}{\partial z} = -g_b(z_*)\frac{F_s}{z_P w_*} - g_t(z_*)\frac{F_e}{z_P w_*}. \tag{5}$$

A first assumption of MLG is mixed-layer similarity to find universal relationships between BL variables (Stull, 2012): 45 $z_* = \frac{z}{z_P}$ is the ratio of the measurement height $z$ and the particle mixing height $z_P$, and $w_*$ is the convective velocity scale or Deardorff velocity. We use the particle mixing height $z_P$ as a proxy for the inversion height $z_i$, which is used in the original MLG method. The particle mixing height 50 $z_P$ was determined using the profiles of the particle number concentration, the potential temperature, absolute humidity, and the liquid water content, and $z_P$ is defined as the height where the gradients of the profiles clearly change. If there are clouds, the height below the cloud layer is used as $z_P$. 55

**Table 1.** Comparison of EC, $K$-theory, and MLG requirements and challenges.

|  | EC | $K$ theory | MLG |
|---|---|---|---|
| Basis of calculation | Eddy covariance | Flux–gradient similarity | Flux–gradient similarity |
| Required data | vertical wind speed $w$, particle number concentration $N$ | particle number concentration $N$, in $\geq 2$ heights | particle number concentration $N$, in $\geq 3$ heights |
| Typical measurement frequency | fast ($\geq 10\,\text{Hz}$) | slow ($> 0.1\,\text{Hz}$) | slow ($> 0.1\,\text{Hz}$) |
| Additional parameters | none | $K$ universal functions | $w_*$, $z_P$, TD-BU functions |
| General conditions | developed turbulence neutral/unstable | neutral stability or universal functions | well-mixed MBL, unstable, neutral |
| Challenges | moving platform, short flight legs | nonlinear gradients | height/concentration uncertainties |

The Deardorff velocity, $w_* = \left( \frac{g}{\Theta_v} \overline{w' \Theta_v'} z_P \right)^{\frac{1}{3}}$, characterizes the turbulent mixing due to free convection, with the gravitational constant $g$, the virtual potential temperature $\Theta_v$, and the buoyancy flux at the surface $\overline{w' \Theta_v'}$.

A second assumption in MLG is that top-down (TD) and bottom-up (BU) transport each obey separate flux–gradient relationships. The top-down and the bottom-up diffusivities are described by two dimensionless analytical functions $g_t$ and $g_b$. In contrast to $K$ theory, here the diffusivity is a function of height. In addition to turbulent exchange, the entrainment flux is influenced by mesoscale variability caused by small clusters of cumulus clouds, variation in horizontal wind or Kelvin–Helmholtz instabilities (Lenschow et al., 1999). Therefore, clear-sky conditions and a horizontal homogeneous surface are assumed. Top-down and bottom-up gradient functions above the ocean modeled by large eddy simulations (LESs) were taken from Moeng and Wyngaard (1989):

$$g_t(z_*) = 0.4(z_*)^{-\frac{3}{2}}, \tag{6a}$$

$$g_b(z_*) = 0.7(1 - z_*)^{-2}. \tag{6b}$$

The gradient is calculated out of two measured concentrations from different heights. Thus, Eq. (5) must be integrated over height between the two heights of concentration measurements normalized with the particle mixing height, $z_{*1}$ and $z_{*2}$. In order to calculate the two unknown fluxes $F_s$ and $F_e$, at least three concentration measurements within the BL are needed to have at least two equations for two different gradients.

For the MLG fluxes calculated in this study, the concentration difference was calculated between three different heights of the median profile of the particle number concentration. These heights were chosen close to the surface and inversion height and in the middle of the MBL profile. The resulting two equations were solved analytically after integration of Eq. (5). In order to calculate $w_*$, a horizontal leg close to the ocean surface is needed. Thus, all available low horizontal flight legs were used to estimate the median value and standard deviation of $w_* = 0.62 \pm 0.17\,\text{m s}^{-1}$.

In order to estimate the uncertainty of the fluxes estimated by the MLG method, MCS was applied, similar to the MCS procedure for $K$ theory. Parameter values were taken randomly from uniform distributions, assuming a $10\,\%$ uncertainty for aerosol particle number concentration ($N \pm 0.1N$), variation of $w_* \pm 0.17\,\text{m s}^{-1}$, and particle mixing height $z_P \pm 50\,\text{m}$.

### 2.3.4 Application and limitations of each method in comparison

All three methods used to estimate vertical particle fluxes in the MBL are suitable for different applications, they have different limitations, uncertainties, and underlying assumptions. Horizontal homogeneity and stationarity are assumed for all of them. Three-dimensional airborne measurements cannot distinguish if variations occur due to temporal variations or spatial inhomogeneities. Fluctuations of particle number concentrations might be caused by turbulent mixing but also by variable sources or sinks such as new particle formation or coagulation. The assumption of horizontal homogeneity and stationarity was applied due to generally low number concentrations, a homogeneous surface below and no obvious sources for aerosol particles. Table 1 summarizes additional requirements and challenges of the three methods. EC requires time series of vertical wind speed and particle number concentration at a reference height. Limited time resolution of the CPC measurement results in a loss of high-frequency flux contributions, which can be spectrally corrected (Horst, 1997). In airborne EC flux measurements from a moving platform, the resolution of turbulent fluctuations is limited by the sampling frequency and the true air speed. In this study, the true air speed of the measurement platform was relatively slow, which is beneficial for the res-

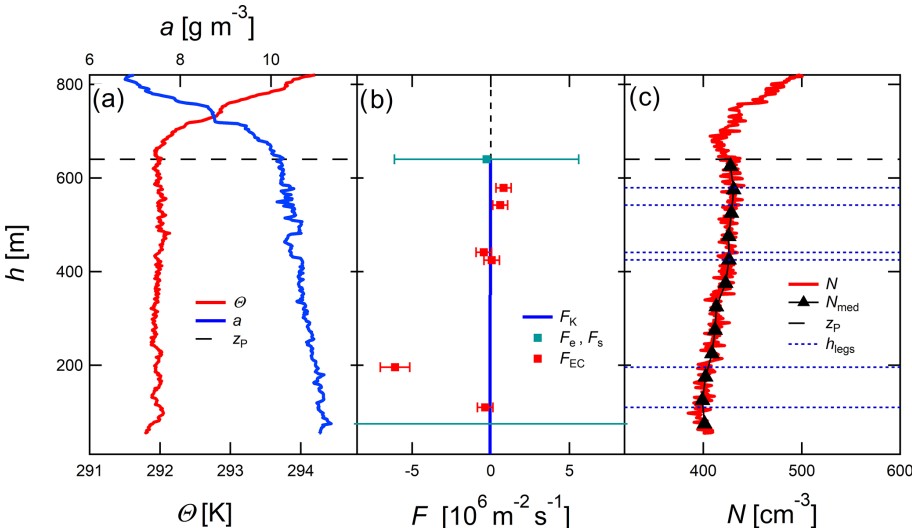

**Figure 3.** Data from flight no. 3, 5 July. **(a)** Vertical profile of potential temperature (red) and absolute humidity (blue). The horizontal dashed black line shows the particle mixing height $z_P$. **(b)** Profile of aerosol particle flux estimates of all three methods including uncertainties of $F_e$, $F_s$, and $F_{EC}$. The particle flux results of the MLG ($F_e$, $F_s$) and $K$ ($F_K$) methods, which are based on sections of the median profiles **(c)**, are shown in blue and green. The estimates of the EC method ($F_{EC}$) are shown in red; they are based on horizontal flight legs shown in **(c)**. **(c)** Profile and median profile of aerosol particle number concentration of the CPC (red and black) as well as the height of the horizontal flight legs (dotted blue lines).

olution of fast fluctuations in the EC method. Furthermore, particle fluxes can be calculated directly by EC without any additional parameter required.

In contrast to that, $K$ theory is using at least two, and MLG at least three, slow concentration measurements at different heights within the MBL and also additional parameters are required. In $K$ theory, the vertical turbulent diffusivity $K$ has to be calculated to estimate the particle flux. For MLG, the particle mixing height $z_P$ and $w_*$ have to be calculated, and the top-down and bottom-up functions have to be determined to estimate the particle surface and entrainment fluxes. $K$ theory as presented in Eq. (2) is applied under neutral conditions, while in the surface layer non-neutral stability conditions can be taken into account with universal functions. The MLG approach is based on mixed-layer scaling and requires a well-mixed MBL. Over the ocean, neutral conditions are typically expected, but stable conditions and weakly unstable conditions may occur. In $K$ theory, nonlinear particle concentration profiles are only conditionally suitable to calculate a vertical gradient. Both in the $K$-theory and the MLG methods, the smaller the vertical gradients of particle number concentration are, the stronger the effect of measurement uncertainties on the flux estimate will be. Finally, it should be noted that EC estimates a particle flux $F_{EC}$ across a reference height, which is the flight leg height in this study. In contrast, the $K$-theory flux estimate $F_K$ represents the profile segment between the concentration measurements used to calculate the gradient, and the MLG method yields two different estimates: (i) the surface flux estimate $F_s$, in this study at the interface between the ocean and the MBL, and study at the interface between the ocean and the MBL, and

(ii) the entrainment flux estimate $F_e$ at the interface between the MBL and the free troposphere. Due to these very different approaches and assumptions, variations between the results of the three methods for the same case study are expected.

## 3 Results and discussion

Aerosol particle flux estimates of the three introduced methods will be shown and discussed focusing on case studies in order to demonstrate the main results and emerging challenges. Research flight nos. 3, 4, 5 and 7 are chosen to highlight the main results. During flight nos. 3, 5 and 7 (see Siebert et al., 2021), the MBL was well mixed, and thus the focus will be put on the comparison between the different methods. Flight no. 4 is chosen to introduce the methods for a case without well-mixed boundary layer conditions, illustrating special features of the profiles and their effects on flux estimates.

### 3.1 Particle flux estimates in well-mixed MBL: comparison of different methods

Figure 2 shows vertical profiles observed in a clear-sky, well-mixed MBL on 10 July 2017. On that day, the particle mixing height $z_P$ was estimated from the temperature and humidity profiles (Fig. 2a) and from the particle number profile (Fig. 2c) to be at 600 m. Within the MBL, three horizontal flight legs were flown at around 130, 275, and 535 m height. The uncertainty of altitude was approximately $\pm 12$ m.

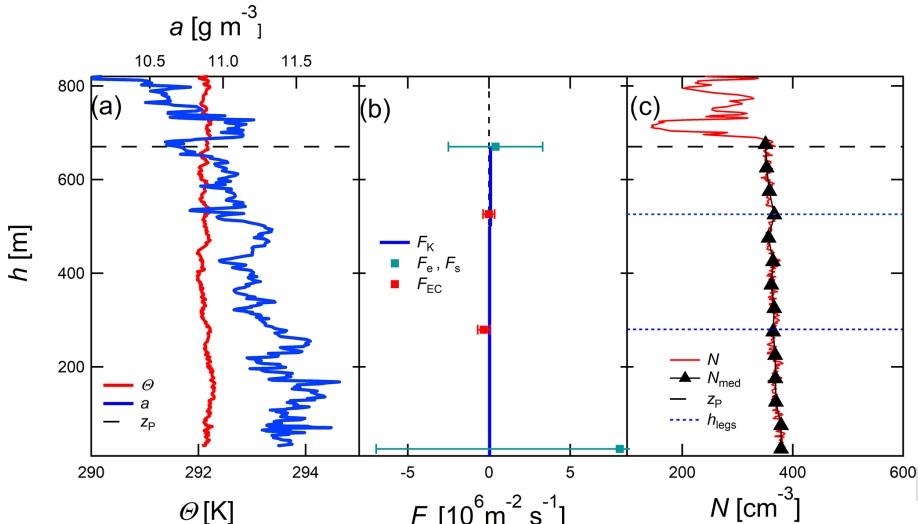

**Figure 4.** Data from flight no. 5, 8 July. **(a)** Vertical profile of potential temperature (red) and absolute humidity (blue). The horizontal dashed black line shows the particle mixing height $z_P$. **(b)** Profile of aerosol particle flux estimates of all three methods including uncertainties of $F_e$, $F_s$, and $F_{EC}$. The particle flux results of the MLG ($F_e$, $F_s$) and K ($F_K$) methods which are based on sections of the median profiles **(c)**, are shown in blue and green. The estimates of the EC method ($F_{EC}$) are shown in red; they are based on horizontal flight legs shown in **(c)**. **(c)** Profile and median profile of aerosol particle number concentration of the CPC (red and black) and the height of the horizontal flight legs (dotted blue lines).

The particle number concentration in the MBL on 10 July 2017 was the highest compared to all other flights, increasing from about 800 at ground level to 1000 cm$^{-3}$ around $z_P$ (see Table 2).

Particle fluxes plus uncertainties of the EC method (Fig. 2b) were calculated from data of horizontal flight legs within the MBL. $K$-theory fluxes and MLG fluxes were calculated with median profile data shown in Fig. 2c. In order to apply $K$ theory, the profile was split into three linear parts
and fluxes for these three different height ranges were calculated. The particle fluxes estimated by the different methods agree well within the range of uncertainties for the MLG entrainment flux $F_e$, $F_K$ in the layer close to the particle mixing height (450–600 m) and $F_{EC}$ in the top segment of the mixing
layer (530 m) (Fig. 2b). $F_{EC}$ represents a local balance at the measurement height, while $F_K$ represents the selected part of the profile while the flux estimated with MLG considers the whole profile.

The MLG surface flux $F_s = -77 \times 10^6$ m$^{-2}$ s$^{-1}$ (off scale
in Fig. 2b) was more than 2 orders of magnitude larger than $F_K$ in the heights $< 300$ m. $F_{EC}$ in the lower part of the mixing layer (130, 275 m) and $F_K$ in the middle and the lower part of the MBL had very small values. Except for $F_{EC}$ in the lowest leg (130 m), the flux direction in the section near the
surface and the section near the inversion was consistent for all different flux calculation methods. The results show that aerosol particle transport in the upper section of MBL was directed upwards into the FT on that day. In the lower part, two out of three methods show a downwards directed particle
flux, i.e., particles deposit in the sea surface.

Most of the uncertainty ranges of the flux estimates passed through zero, which means that in these cases even the sign of the flux cannot be unambiguously determined. Uncertainty ranges of the fluxes due to counting statistics were calculated following Buzorius et al. (2003) and Fairall (1984) for the 35 EC fluxes, and by MCS for the fluxes estimated by $K$ theory and MLG. The random flux uncertainty due to limited particle counting statistics was estimated to range between 0.1 and $0.8 \times 10^6$ m$^{-2}$ s$^{-1}$, which is in the same order of magnitude as most flux estimates. However, with the random shuf- 40 fle method by Billesbach (2011) it could be shown that 15 of 21 EC fluxes presented in Table 3 are larger than the 95 % confidence interval of the flux contribution of random instrument noise.

On 5 July 2017, the profile of flight no. 3 was flown at 45 around 14:30 LT (local time) for 15 min (Fig. 3) followed by six horizontal legs within the MBL. The MBL was well mixed and a cloud coverage of 2/8 was observed due to a few small cumulus clouds.

On that day, fluxes estimated by the gradient methods had 50 very small values (below $10^5$ m$^{-2}$ s$^{-1}$), which was expected due to the weak gradient within the MBL. The uncertainty ranges resulting from the measurements show that the direction of the flux was again not clear. Stronger gradients would result in more robust results of the gradient methods. Flux es- 55 timates calculated by EC confirmed the small net exchange of particles on this day. The surface flux estimated by the MLG method was strong again and directed downwards with a value of $-18.8 \times 10^6$ m$^{-2}$ s$^{-1}$. At the same time, the uncertainty ranges were larger than the flux estimates, indicating 60

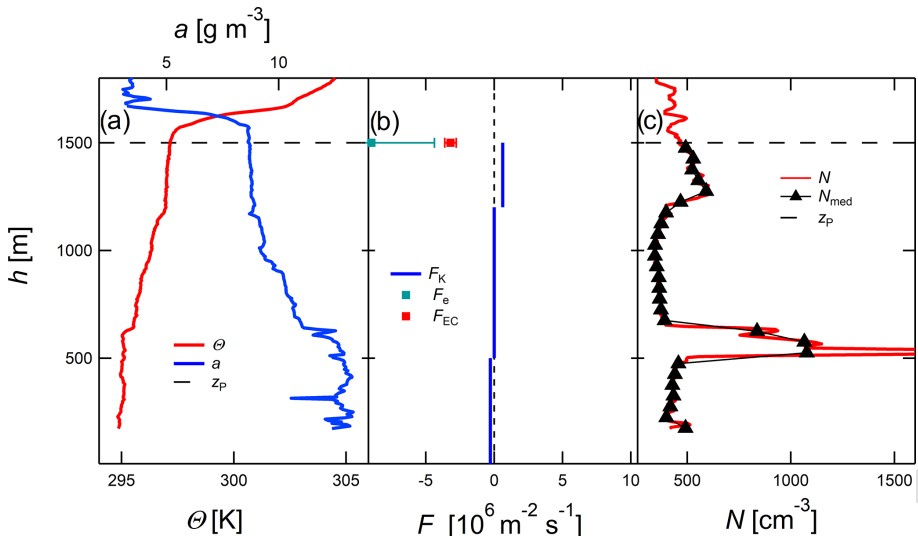

**Figure 5.** Data from flight no. 4, 7 July. **(a)** Vertical profile of potential temperature (red) and absolute humidity (blue). The horizontal dashed black line shows the particle mixing height $z_P$. **(b)** Profile of aerosol particle flux estimates of all three methods including uncertainties of $F_e$ and $F_{EC}$. The particle flux results of the MLG ($F_e$, $F_s$) and $K$ ($F_K$) methods, which are based on sections of the median profiles **(c)**, are shown in blue and green. The estimate of the EC method ($F_{EC}$) is shown in red; this is based on one horizontal flight leg in the height of $z_P$. **(c)** Profile and median profile of aerosol particle number concentration of the CPC (red and black) and the height of the horizontal flight legs (dotted blue lines).

a very large uncertainty of the surface flux estimated by the MLG method.

The profile of flight no. 5 on 8 July 2017 (Fig. 4) started at 14:45 LT and took 17 min. The particle mixing height was identified at $z_P = 670$ m, and according to $\partial_z \Theta \approx 0$ the MBL was well-mixed. The conditions were similar to the conditions of flight no. 3 (Fig. 3), but a layer with 4/8 stratocumulus has been developed. The well-mixed layer, i.e., the layer with nearly constant values in $N$, ends within the cloud base, explaining the drop of particle concentration in the upper part of the profile (Fig. 4c). Just below $z_P$, a weak particle concentration gradient is visible, and consequently a slightly positive but very small $F_K$ is estimated. The entrainment flux estimated by MLG shows the same tendency.

In all case studies shown here (flight nos. 3, 5, 7) and also on other days, fluxes estimated by $K$ theory and EC in the upper part of the MBL were comparable. In addition, $F_e$ typically agreed with the flux estimates in that height within the range of uncertainty.

The surface fluxes estimated by the MLG method were in all cases much larger than the other estimated fluxes in the surface layer (e.g., flight nos. 3 and 7), where the strongest gradients are expected due to the interface between ocean and atmosphere. Fluxes calculated according to the EC or $K$ theory were not determined as close to the surface. On the other hand, the MLG surface fluxes often seem to be too large to be plausible. One reason for this could be that near-surface flights to determine the gradient were not possible for safety reasons. This could be one source of uncertainty for $F_s$.

## 3.2 Particle flux estimates with complex aerosol layering

The cases shown in Sect. 3.1 are based on aerosol concentration profiles with more or less monotonic gradients. However, estimating particle fluxes becomes more challenging for situations with more complex aerosol layering, as shown in Fig. 5 for flight no. 4 on 7 July 2017. The particle mixing height $z_P = 1500$ m was located at the cloud base. During the flight, stratocumulus clouds were dissipating over the ocean, while some isolated convective cumulus clouds were observed close to the island. Vertical profiles of particle number concentration were highly variable with an aerosol concentration peak above 500 m (Fig. 5c). In addition, the potential temperature profile changed at this height (Fig. 5a), indicating a decoupling between the surface and the sub-cloud layer.

Fluxes estimated by MLG and $K$ theory near the inversion were opposite in direction, and thus the results of the different methods were not comparable. $F_{EC}$ close to $z_P$ showed results comparable to $F_e$ estimated by the MLG method. For variable profiles, MLG is highly uncertain or even not applicable since the top-down and bottom-up functions are fixed, while $K$ theory can be adapted to the profile by choosing linear parts of the profile.

One possible reason for profiles like the one shown in Fig. 5 are decoupled layers (Dong et al., 2015) within the MBL where different air masses lie on top of each other.

**Table 2.** Overview of all profiles where flux estimation methods were applied: start time and duration of profile, particle mixing height $z_P$, mean and standard deviation of aerosol particle number concentration $N$, and the mean and standard deviation of potential temperature $\Theta$ within the whole MBL profile and the cloud properties. An overview over the synoptic situation, more meteorological parameters, and their vertical profiles can be found in Siebert et al. (2021). Flight nos. 3, 4, 5, and 7, which were selected for discussion, are highlighted in bold.

| Date | Flight | Profile | Start time [h] | Duration [min] | $z_P$ [m] | $N$ [cm$^{-3}$] | $\Theta$ [K] | Clouds* |
|---|---|---|---|---|---|---|---|---|
| 4 Jul 2017 | no. 2 | 1 | 13.35 | 10.39 | 1400 | $560 \pm 16$ | $294.4 \pm 1.4$ | thin Sc |
| 4 Jul 2017 | no. 2 | 2 | 14.63 | 7.53 | 1000 | $517 \pm 43$ | $293.9 \pm 0.9$ | thin Sc |
| **5 Jul 2017** | **no. 3** | **1** | **14.53** | **15.84** | **640** | **$416 \pm 12$** | **$291.8 \pm 0.2$** | **few Cu** |
| 7 Jul 2017 | no. 4 | 1 | 10.65 | 18.45 | 1500 | $534 \pm 199$ | $295.8 \pm 0.7$ | dissip. Sc, convective Cu |
| **7 Jul 2017** | **no. 4** | **2** | **12.08** | **5.83** | **1500** | **$491 \pm 197$** | **$295.6 \pm 0.5$** | **dissip. Sc, convective Cu** |
| **8 Jul 2017** | **no. 5** | **1** | **14.74** | **17.33** | **670** | **$365 \pm 8$** | **$292.2 \pm 0.3$** | **low Sc** |
| 9 Jul 2017 | no. 6 | 1 | 9.77 | 9.71 | 1050 | $590 \pm 9$ | $291.8 \pm 0.2$ | thick Sc |
| 9 Jul 2017 | no. 6 | 2 | 11.18 | 6.46 | 1050 | $532 \pm 8$ | $291.8 \pm 0.1$ | thick Sc |
| **10 Jul 2017** | **no. 7** | **1** | **11.20** | **13.64** | **600** | **$913 \pm 38$** | **$292.3 \pm 0.1$** | **only few Cu hum** |
| 13 Jul 2017 | no. 8 | 1 | 14.25 | 12.07 | 1250 | $330 \pm 24$ | $296.9 \pm 0.9$ | Sc |
| 14 Jul 2017 | no. 9 | 1 | 13.78 | 18.42 | 1200 | $273 \pm 31$ | $295.8 \pm 1.2$ | Sc |
| 15 Jul 2017 | no. 10 | 1 | 14.97 | 11.78 | 1000 | $160 \pm 59$ | $296.4 \pm 0.6$ | several St/Sc layers, few Cu |
| 16 Jul 2017 | no. 11 | 1 | 10.02 | 14.95 | 1000 | $134 \pm 13$ | $295.5 \pm 0.9$ | Sc |
| 16 Jul 2017 | no. 12 | 1 | 14.36 | 9.10 | 850 | $207 \pm 78$ | $296.4 \pm 0.4$ | few Cu below Sc layer |
| 18 Jul 2017 | no. 14 | 1 | 16.86 | 5.00 | 730 | $193 \pm 37$ | $293.7 \pm 0.3$ | quite homogeneous Sc |
| 21 Jul 2017 | no. 15 | 1 | 10.03 | 5.00 | 800 | $400 \pm 5$ | $293.8 \pm 0.1$ | Sc |
| 21 Jul 2017 | no. 15 | 2 | 11.26 | 9.17 | 1200 | $445 \pm 49$ | $293.8 \pm 0.3$ | Sc |
| 21 Jul 2017 | no. 16 | 1 | 14.53 | 7.83 | 1280 | $459 \pm 27$ | $294.2 \pm 0.3$ | thin, dissip. Sc, Sc layer above |

* Taken from Siebert et al. (2021): Sc stands for stratocumulus, Cu stands for cumulus, and Cu hum stands for cumulus humilis.

## 3.3 Overview of particle flux results

Characteristic parameters of the 18 profiles flown during all research flights with the helicopter-borne platform ACTOS are shown in Table 2. In order to calculate and interpret these fluxes, the start time, the duration of the profile, and the particle mixing height $z_P$ are important. Mean and standard deviation of aerosol particle number concentration and the mean and standard deviation of potential temperature within the whole MBL profile are useful to characterize different profiles and to assess the environmental conditions. A large standard deviation of the particle number concentration might be caused by strong gradients within the MBL or by layers with particle concentration peaks due to poor mixing (e.g., flight no. 4).

An overview of particle fluxes and uncertainties estimated by all three methods is given in Table 3 for all 18 profiles. Thus, the flux estimates can be compared for individual profiles but also between the methods in general. For $K$ theory, three fluxes are given: first, the whole profile of the MBL is used for the flux calculation ($F_{K,\mathrm{MBL}}$), and then, if the profile is split up, the lowest and the highest parts of the MBL profile are used ($F_{K,\mathrm{bottom}}$ and $F_{K,\mathrm{top}}$). This distinction is also a way to check if the chosen splitting of the profiles is reasonable. For flight no. 7, for example, $F_{K,\mathrm{MBL}}$ was very different from $F_{K,\mathrm{bottom}}$ and $F_{K,\mathrm{top}}$. For comparison of fluxes estimated by MLG close to the surface ($F_s$) and close to the entrainment zone ($F_e$), $F_K$ and $F_{EC}$ in the lowest and highest parts of the

MBL should be considered. For EC, only the flux estimates calculated from the lowest and the highest flight legs within the MBL are given ($F_{EC,\mathrm{bottom}}$ and $F_{EC,\mathrm{top}}$). The altitude of the flight leg is given in brackets, and NA means there was no horizontal flight leg in that region.

We report typical particle number fluxes of $10^4$–$10^6$ m$^{-2}$ s$^{-1}$. This is several orders of magnitude lower than urban particle number fluxes. Typical urban particle number fluxes measured by eddy covariance with CPCs are up to $10^6$ m$^{-2}$ s$^{-1}$, e.g., in Manchester, London, Edinburgh, Gothenburg (Martin et al., 2009), and $0.9 \times 10^9$ m$^{-2}$ s$^{-1}$ in Edinburgh (Dorsey et al., 2002). Kurppa et al. (2015) report a median value of $0.18 \times 10^9$ m$^{-2}$ s$^{-1}$ in Helsinki, and Conte et al. (2021) report median values of $0.21 \times 10^9$ m$^{-2}$ s$^{-1}$ in Lecce and $0.04 \times 10^9$ m$^{-2}$ s$^{-1}$ in Innsbruck.

In non-urban areas, typical aerosol number fluxes above tall vegetation are up to $0.1$–$0.2 \times 10^9$ m$^{-2}$ s$^{-1}$ (e.g., Buzorius et al., 2000; Held and Klemm, 2006). Flanagan et al. (2005) report particle number fluxes of the order of $10^9$–$10^{10}$ m$^{-2}$ s$^{-1}$ during nucleation events at the Irish Atlantic coastline. In contrast, particle number fluxes observed in the Arctic Ocean are 1 to 2 orders of magnitude smaller than the fluxes reported in this study.

Nilsson and Rannik (2001) report median particle number fluxes of $1 \times 10^4$ m$^{-2}$ s$^{-1}$ above open leads and ice floes and $25 \times 10^4$ m$^{-2}$ s$^{-1}$ above the open sea. Held et al. (2011) re-

**Table 3.** Overview of the aerosol fluxes and their uncertainties for each profile of the CPC estimated by all three methods: $K$ theory, MLG, and EC. The unit of all fluxes is $10^6$ m$^{-2}$ s$^{-1}$. Only the lowest and highest flux within the MBL estimated by the EC method is given here. If the EC flux is marked with an asterisk (*), it is not possible to distinguish whether it is a random or actual flux (Billesbach, 2011). The altitude of the flight leg is given in brackets, and NA means there was no horizontal flight leg in that region. Flight nos. 3, 4, 5, and 7, which were selected for discussion, are highlighted in bold.

| Date | Flight | Profile | $F_{K,\mathrm{MBL}}$ | $F_{K,\mathrm{bottom}}$ | $F_{K,\mathrm{top}}$ | $F_\mathrm{s}$ | $F_\mathrm{e}$ | $F_{\mathrm{EC,bottom}}$ | $F_{\mathrm{EC,top}}$ |
|---|---|---|---|---|---|---|---|---|---|
| 4 Jul 2017 | no. 2 | 1 | 0.02 ± 0.02 | 0.07 ± 0.05 | −0.09 ± 0.12 | 22.1 ± 25.8 | −1.3 ± 3.1 | 0.6 ± 0.6 (123 m) | 7.8 ± 0.4 (1322 m) |
| 4 Jul 2017 | no. 2 | 2 | 0.12 ± 0.05 | 0.05 ± 0.2 | −0.12 ± 0.12 | 40.5 ± 36.8 | −0.5 ± 4.5 | 0.6 ± 0.6 (123 m) | NA |
| **5 Jul 2017** | **no. 3** | **1** | **−0.05 ± 0.04** | **−0.05 ± 0.1** | **−0.02 ± 0.1** | **−18.8 ± 31.5** | **−0.3 ± 5.9** | **−0.4 ± 0.5 (110 m)** | **0.8 ± 0.5 (579 m)** |
| 7 Jul 2017 | no. 4 | 1 | 0.3 ± 0.11 | −3.93 ± 1.51 | 0.24 ± 0.09 | 101.9 ± 26.2 | −5.8 ± 3.4 | NA | −3.2 ± 0.4* (1500 m) |
| **7 Jul 2017** | **no. 4** | **2** | **0.08 ± 0.04** | **−0.27 ± 0.2** | **0.63 ± 0.38** | **29.7 ± 20.9** | **−9 ± 4.6** | **NA** | **−3.2 ± 0.4* (1500 m)** |
| **8 Jul 2017** | **no. 5** | **1** | **0.03 ± 0.03** | **0.04 ± 0.05** | **0.09 ± 0.18** | **8.1 ± 15.3** | **0.4 ± 2.9** | **−0.3 ± 0.3 (280 m)** | **0 ± 0.3* (526 m)** |
| 9 Jul 2017 | no. 6 | 1 | −0.02 ± 0.02 | −0.02 ± 0.05 | −0.03 ± 0.11 | −5.8 ± 35 | −0.3 ± 2.1 | NA | NA |
| 9 Jul 2017 | no. 6 | 2 | 0.03 ± 0.03 | 0.03 ± 0.04 | 0.06 ± 0.25 | 5.1 ± 14.2 | 0.1 ± 1.6 | NA | NA |
| **10 Jul 2017** | **no. 7** | **1** | **−0.16 ± 0.1** | **−0.35 ± 0.31** | **0.29 ± 0.67** | **−77.1 ± 84.3** | **2.2 ± 7.6** | **0.1 ± 0.8 (129 m)** | **0.7 ± 0.8 (535 m)** |
| 13 Jul 2017 | no. 8 | 1 | 0.03 ± 0.02 | −0.15 ± 0.08 | −0.02 ± 0.03 | 12 ± 20.9 | −1.4 ± 2.2 | NA | NA |
| 14 Jul 2017 | no. 9 | 1 | 0.01 ± 0.01 | −0.03 ± 0.12 | 0.01 ± 0.04 | 46.3 ± 13.6 | −1.7 ± 2.1 | 1 ± 0.3 (97 m) | 0 ± 0.3* (1101 m) |
| 15 Jul 2017 | no. 10 | 1 | 1.8 ± 0.7 | −0.2 ± 0.8 | 1.7 ± 0.6 | 47.1 ± 12 | 1.3 ± 1.5 | NA | NA |
| 16 Jul 2017 | no. 11 | 1 | 0.01 ± 0.01 | −0.07 ± 0.05 | −0.2 ± 0.08 | −1.8 ± 6.5 | −0.7 ± 1.2 | NA | −0.1 ± 0.1 (964 m) |
| 16 Jul 2017 | no. 12 | 1 | −0.34 ± 0.13 | −0.12 ± 0.17 | −0.24 ± 0.18 | −85.8 ± 64.6 | −3 ± 4.2 | NA | NA |
| 18 Jul 2017 | no. 14 | 1 | 0.26 ± 0.1 | 0.07 ± 0.1 | 0.22 ± 0.1 | 64.9 ± 26.8 | 0.2 ± 1.1 | 0.2 ± 0.2 (117 m) | −1.5 ± 0.2 (422 m) |
| 21 Jul 2017 | no. 15 | 1 | 0 ± 0.3 | 0.8 ± 1.9 | 0.3 ± 2 | 6.3 ± 34.5 | −0.1 ± 2.1 | −0.8 ± 0.4 (117 m) | 0 ± 0.4* (431 m) |
| 21 Jul 2017 | no. 15 | 2 | 0.13 ± 0.05 | 0.16 ± 0.21 | 0.04 ± 0.04 | 40.5 ± 18.5 | 0.1 ± 1.2 | −0.8 ± 0.4 (117 m) | 0 ± 0.4* (431 m) |
| 21 Jul 2017 | no. 16 | 1 | 0.05 ± 0.03 | −0.1 ± 0.07 | 0.1 ± 0.07 | −67.2 ± 63.4 | 0.5 ± 2.5 | 1.3 ± 0.4 (425 m) | NA |

port particle number fluxes up to $3 \times 10^4 \, \mathrm{m^{-2} \, s^{-1}}$ above open leads and ice floes in the central Arctic Ocean.

The estimated fluxes were furthermore compared with the dry deposition flux $F_{\mathrm{dry}}$ using the approach $F_{\mathrm{dry}} = -v_{\mathrm{dry}} \cdot N$ (in $\mathrm{cm^{-2} \, s^{-1}}$). From Emerson et al. (2020), for 100 nm particles one can estimate a dry deposition velocity to water in the range of $v_{\mathrm{dry}} = 0.01$ to $0.2 \, \mathrm{cm \, s^{-1}}$. For flight no. 3 on 5 July, the particle number concentration was about $N = 400 \, \mathrm{cm^{-3}}$ at sea surface level, leading to a dry deposition flux $F_{\mathrm{dry}} = -4$ to $-80 \, \mathrm{cm^{-2} \, s^{-1}} = -0.04$ to $-0.8 \times 10^6 \, \mathrm{m^{-2} \, s^{-1}}$. On that day, the EC and $K$ theory flux estimates close to the surface are within this dry deposition flux range, i.e., $F_{\mathrm{EC,bottom}} = -0.4 \times 10^6 \, \mathrm{m^{-2} \, s^{-1}}$ and $F_{\mathrm{K,bottom}} = -0.05 \times 10^6 \, \mathrm{m^{-2} \, s^{-1}}$. The surface flux estimated by MLG, $F_{\mathrm{s}} = -18.8 \times 10^6 \, \mathrm{m^{-2} \, s^{-1}}$, is about 25 times higher compared to the higher estimate. The entrainment flux $F_{\mathrm{e}} = -0.3 \times 10^6 \, \mathrm{m^{-2} \, s^{-1}}$ and the fluxes close to the top of the MBL are in the same order of magnitude.

## 4 Summary and conclusions

Helicopter-borne measurements allow to quantify the vertical exchange of aerosol particles in the MBL by different methods. In this study above the ocean in the Azores region, particle fluxes estimated by EC, $K$ theory, and MLG agreed reasonably well in the upper part of the MBL, while flux estimates close to the surface differed considerably between the methods.

In this study, the observed particle fluxes at the top of the MBL ranged up to $10 \times 10^6 \, \mathrm{m^{-2} \, s^{-1}}$ both in the upward and the downward direction, but most flux values were significantly smaller. In order to illustrate the magnitude of this flux, assuming a well-mixed MBL with a mixing height of 1000 m, a net entrainment flux of $F_{\mathrm{e}} = 10 \times 10^6 \, \mathrm{m^{-2} \, s^{-1}}$ would change the particle number concentration in the MBL by 30 to $40 \, \mathrm{cm^{-3} \, h^{-1}}$. In many cases, the entrainment flux $F_{\mathrm{e}}$ of the MLG method agreed within the range of uncertainty with $F_{\mathrm{EC}}$ and $F_K$ estimates close to the top of the MBL. This suggests that all three methods can be applied to estimate the net particle exchange at the interface between the MBL and the FT, depending on the flight track with respect to number, height and length of horizontal flight legs or profiles within the MBL.

When comparing these different results, the main differences between the methods must also be taken into account. In order to quantify the net particle exchange between MBL and FT, the EC method requires a horizontal flight leg at the top of the MBL, while $K$ theory would extrapolate a profile measurement at the top of the MBL. For the calculation of the entrainment flux by the MLG method, concentration measurements at three different heights across the mixing layer are required.

For this study, observations close to the surface are not available, which increases the uncertainties of the surface flux estimates of the MLG method. $F_{\mathrm{s}}$ was typically much larger and in most cases unrealistically high compared to $F_{\mathrm{EC}}$ and $F_{\mathrm{K}}$ close to the surface.

$K$-theory and MLG flux estimates are less sensitive to the selection of data from different heights if the MBL is well mixed; however, the flux estimates are more robust for strong gradients. Fast measurements of vertical wind speed and particle number concentration and a relatively slow flight speed are beneficial to cover the entire turbulence spectrum when using EC. However, low particle number concentrations above the ocean cause poor counting statistics, which also increase uncertainties in particle number concentration gradients for the $K$-theory and MLG methods. A CPC with a larger sample flow rate would decrease the error due to sampling statistics.

It is undisputed that the uncertainties in all three measurement methods are still quite large. Nevertheless, the results of this study contribute to a better understanding of the particle transport between MBL and FT and the distribution of particles within the MBL. In particular, they show the fundamental problems that still exist in flux determination despite the fact that the helicopter-borne ACTOS provides a slow-flying platform that minimizes the basic degradation of both turbulence and aerosol measurements compared to fast-flying aircraft.

A promising approach for a more robust measurement of particle flux with the EC method would be a faster CPC as described in Wehner et al. (2011) combined with a significantly increased volume flux to minimize statistical uncertainty. The latter is especially fundamentally important in environments with comparably lower particle number concentrations such as the Azores or Polar regions.

**Data availability.** Datasets used in the current study are available from the corresponding author on request.

**Author contributions.** JL, HS, and BW performed the helicopter-borne measurements. JL, AH, and BW analyzed the aerosol data. HS processed and checked the wind data. MM prepared data for EC analysis and also did the analysis. JL estimated and compared the fluxes with the three methods. AH and BW discussed and evaluated the results together with JL. JL prepared the manuscript, and all authors contributed to the paper writing and edited the text.

**Competing interests.** The contact author has declared that none of the authors has any competing interests.

**Acknowledgements.** This project was supported by several grants of the Deutsche Forschungsgesellschaft (DFG, with grant nos. SI 1543/4-1, WE 1900/33-1, WE 2757/2-1, and HE 6770/2-1). We had the possibility to stay at Graciosa Airport during the measurements. Thank you for the great support to Rui Medeiros, Maria Manuela Santos, and the whole SATA Air Açores team. For the helicopter operation we thank the two pilots Alwin Vollmer and Jürgen Schütz from Rotorflug GmbH, Germany. Technical support was provided by Astrid Hoffmann, Ralf Kaethner, and Sebastian Duesing (TROPOS). We also thank all participants of the ACORES2017 Campaign, especially the "Aerosol team" of Silvia Henning and Kairne Chevalier (TROPOS).

**Financial support.** This research has been supported by the Deutsche Forschungsgemeinschaft (grant nos. WE 2757/2-1, SI 1543/4-1). TS2

The publication of this article was funded by the Open Access Fund of the Leibniz Association.

**Review statement.** This paper was edited by Leiming Zhang and reviewed by three anonymous referees.

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

## Remarks from the typesetter