# Peer review of "Vertical aerosol particle exchange in the marine boundary layer estimated from helicopter-borne measurements in the Azores region"

_Atmospheric Chemistry and Physics, 2021_

## Author Comment (AC1)

**Reply to Review 1 (RC1)**

The paper reports a study of vertical aerosol fluxes and vertical concentration profiles in the marine boundary-layer. Three different approaches are used to estimate fluxes using helicopter-based measurements and results are compared. I believe that the topic is interesting and there are elements of innovation especially because the MBL is not frequently studied with these approaches and available information is limited in current scientific literature. I also believe that even if the different methods have relatively large uncertainties, results could give useful insights in the exchanges of particles in the marine boundary layer. The topic is suitable for the Journal.

A few aspects should be made more clear in a revision step as mentioned below.

*We thank the reviewer for his comments and suggestions. We reply below to each point in italic font.*

Lines 50-57. Probably it could also be mentioned the work at Mace Head regarding fluxes focused on sea spray.

*We agree that valuable studies regarding sea spray fluxes have been carried out at Mace Head. In the revised version, we add references to the work by Geever et al (2005), de Leeuw et al. (2011) and Ovadnevaite et al. (2014) in the list of examples.*

*We added the following sentence: 'Results from sea spray emission studies are published in e.g., Geever et al. (2005); de Leeuw et al. (2011) and Ovadnevaite et al. (2014).'*

*And these references:*

*Geever, M. C., D. O'Dowd, S. vanEkeren, R. Flanagan, E. D. Nilsson, G. deLeeuw, and Ü. Rannik (2005), Submicron sea spray fluxes, Geophys. Res. Lett., 32, L15810, doi:10.1029/2005GL023081.*

*de Leeuw, G., E. L Andreas, M. D. Anguelova, C. W. Fairall, E. R. Lewis, C. O'Dowd, M. Schulz, and S. E. Schwartz (2011), Production flux of sea spray aerosol, Rev. Geophys., 49, RG2001, doi:10.1029/2010RG000349.*

*Ovadnevaite, J., Manders, A., de Leeuw, G., Ceburnis, D., Monahan, C., Partanen, A.-I., Korhonen, H., and O'Dowd, C. D.: A sea spray aerosol flux parameterization encapsulating wave state, Atmos. Chem. Phys., 14, 1837–1852, https://doi.org/10.5194/acp-14-1837-2014, 2014.*

Line 99. Better to say less demanding or less difficult rather than less serious.

*Changed to 'less demanding'.*

There is a confusion of time resolution (that should be in s) and frequency (in Hz), for example in line 112 or in Table 1.

*We agree, thank you for pointing out the inconsistency use of these terms. We make the following changes:*

*Line 73: "limited time resolution" instead of "low time resolution"*

*Line 113: "time resolution of 0.1 s" instead of "time resolution of 10 Hz"*

*Line 136: "with high measurement frequency" instead of "at high time resolution"*

*Line 146: "with 1 s time resolution" instead of "with 1 Hz resolution"*

*Table 1 (third entry): "Typical measurement frequency" instead of "Time resolution"*

In line 112 there is just the difference between sampling frequency (10 Hz) and time resolution of the instrument (1 s or 1 Hz). That means we sampled faster than the real resolution of the instrument is.

*See above.*

Line 113. How have been done the correction for aerosol losses? For this is usually necessary to have a measurement of size distribution.

*Yes, particle losses depend on the particle size. We measured the size resolved particle losses through the diffusion dryer and calculated the losses through the inlet system. The result of both was a size-dependent function of particle losses, which can of course not be applied to a CPC measurement of total particle number concentration. Thus, the mean value for particles between 20 and 100 nm was chosen and applied to the data.*

*We modified the sentence to: 'Aerosol number concentrations are corrected for losses in the inlet system using a mean factor for diameters between 20 and 1000 nm that has been determined experimentally and for variations in the sample flow due to pressure changes.'*

Lines 116-119. This pendulum motion was seen on meteorological measurements?

*Yes, it was visible in the wind measurements of all three coordinates, and therefore treated with a spectral band-stop filter as mentioned in the manuscript.*

It could be useful to discuss how the magnitude of fluxes compare with measurements in different environments that could help the reader to make more sense of the large uncertainties and of the role of counting errors. They seems to be significantly lower than those observed in urban areas but likely larger or comparable with those observed in polar regions.

*We report typical particle number fluxes of $10^4$ - $10^6$ $m^{-2}$ $s^{-1}$. This is several orders of magnitude lower than urban particle number fluxes. Typical urban particle number fluxes measured by eddy covariance with CPCs are up to $10^9$ $m^{-2}$ $s^{-1}$, e.g. in Manchester, London, Edinburgh, Gothenburg (Martin et al. 2009), 0.9 x $10^9$ $m^{-2}$ $s^{-1}$ in Edinburgh (Dorsey et al. 2002). Kurppa et al. (2015) report a median value of 0.18 x $10^9$ $m^{-2}$ $s^{-1}$ in Helsinki, and Conte et al. (2021) report median values of 0.21 x $10^9$ $m^{-2}$ $s^{-1}$ in Lecce and 0.04 x $10^9$ $m^{-2}$ $s^{-1}$ in Innsbruck.*

*In non-urban areas, typical aerosol number fluxes above tall vegetation are up to 0.1 – 0.2 x $10^9$ $m^{-2}$ $s^{-1}$ (e.g. Buzorius et al. 2000; Held and Klemm 2006). Flanagan et al. (2005) report*

*particle number fluxes of the order of $10^9$ to $10^{10}$ $m^{-2}$ $s^{-1}$ during nucleation events at the Irish Atlantic coastline.*

*In contrast, particle number fluxes observed in the Arctic Ocean are one to two orders of magnitude smaller than the fluxes reported in this study. Nilsson and Rannik (2001) report median particle number fluxes of 1 x $10^4$ $m^{-2}$ $s^{-1}$ above open leads and ice floes, and 25 x $10^4$ $m^{-2}$ $s^{-1}$ above the open sea. Held et al. (2011) report particle number fluxes up to 3 x $10^4$ $m^{-2}$ $s^{-1}$ above open leads and ice floes in the Central Arctic Ocean.*

*Buzorius, G., Rannik, Ü., Mäkelä, J.M., Keronen, P., Vesala, T., Kulmala, M., 2000. Vertical aerosol fluxes measured by the eddy covariance method and deposition of nucleation mode particles above a Scots pine forest in southern Finland. Journal of Geophysical Research 105, 19905–19916.*

*Conte, M., Contini, D., Held, A., 2021. Multiresolution decomposition and wavelet analysis of urban aerosol fluxes in Italy and Austria. Atmospheric Research 248, 105267. doi.org/10.1016/j.atmosres.2020.105267*

*Dorsey, J.R., Nemitz, E., Gallagher, M.W., Fowler, D., Williams, P.I., Bower, K.N., Beswick, K.M., 2002. Direct measurements and parameterisation of aerosol flux, concentration and emission velocity above a city. Atmospheric Environment 36, 791–800.*

*Flanagan, R.J., Geever, M., O'Dowd, C.D., 2005. Direct Measurements of new-particle fluxes in the coastal environment. Environ. Chem., 2005, 2, 256–259.*

*Held, A. and Klemm, O., 2006. Direct measurement of turbulent particle exchange with a twin CPC eddy covariance system. Atmos. Environ. 40, S92-102.*

*Held, A., Brooks, I.M., Leck, C., and Tjernström, M., 2011. On the potential contribution of open lead particle emissions to the central Arctic aerosol concentration. Atmos. Chem. Phys. 11, 3093-3105.*

*Kurppa, M., Nordbo, A., Haapanala, S., Järvi, L., 2015. Effect of seasonal variability and land use on particle number and CO2 exchange in Helsinki, Finland. Urban Climate 13, 94-109. https://doi.org/10.1016/j.uclim.2015.07.006*

*Martin, C.L., Longley, I.D., Dorsey, J.R., Thomas, R.M., Gallagher, M.W., Nemitz, E., 2009. Ultrafine particle fluxes above four major European cities. Atmospheric Environment 43 , 4714-4721. https://doi.org/10.1016/j.atmosenv.2008.10.009*

*Nilsson, E.D., Rannik, Ü., 2001. Turbulent aerosol fluxes over the Arctic Ocean: 1. Dry deposition over sea and pack ice. Journal of Geophysical Research 106, 32125–32137.*

Median values are used instead of averages for gradient and MLG approach. Is there a reason? I mean did authors verified that it is better compared to the more widely used average values?

*In a few cases, extreme values that may bias the arithmetic mean values occurred. Therefore, we used the more robust median values.*

Line 223. Better airborne.

*Changed to airborne.*

Caption of Table 2. It is needed a subscript in zP.

*Thanks for the hint, we changed it accordingly.*

Considering that uncertainties are often quite large and in several instances also the sign of flux could be ambiguous, it would be useful an effort to summarise in the conclusions what can be concluded and what needs further studies regarding particle exchanges in the MBL. It would also be useful to conclude, if possible, what is the more suitable calculation approach for fluxes in the conditions studied.

*It is not possible to conclude from this study what is the most suitable calculation approach. The only conclusion is written in the coclusion section: Observed entrainment flux could supply in the order of 30-40 particles/cm3 per hour to the MBL.*

---

## Author Comment (AC2)

**Reply to Review 3 (RC2):**

The authors present a novel analysis of airborne (helicopter-based) vertical fluxes of aerosol particle number concentrations. Three separate techniques for deriving vertical fluxes are explored and a systematic discussion of their strengths and weaknesses are included. The authors present a fair assessment of the limitations of the techniques which will be valuable for future analyses. The paper focuses primarily on measurements of the entrainment flux of aerosol from the free troposphere, concluding that in the airmasses sampled here, entrainment could supply 30-40 particles/cm3 per hour to the MBL.

My only comment is that it would be helpful to expand on this last point a bit more to include a short discussion on the sources and sinks of particles in the MBL and the extent to which numbers of this magnitude (30 p/cm3 h) compare with what one might estimate for dry deposition to the ocean surface or that needed to sustain some of the larger NPF events that have been sampled at ENA. This might help the reader (and future scientists) get a better handle to the limitations of this approach in the context of the magnitude of the fluxes required to change particle concentrations in the MBL.

*Thanks for the comment! Below, we included a short discussion here and also to the manuscript.*

*A simple way to estimate dry deposition to the ocean surface is multiplying the particle number concentration N (in $cm^{-3}$) at the surface with the dry deposition velocity $v\_dry$ (in cm $s^{-1}$), i.e. dry deposition flux $F\_dry = - v\_dry * N$ (in $cm^{-2} s^{-1}$)*

*From Emerson et al. (2020, PNAS), for 100 nm particles one can estimate a dry deposition velocity to water in the range of $v\_dry = 0.01$ cm/s to 0.2 cm/s.*

*For flight #3 on July 5$^{th}$, we find a particle number concentration of about $N = 400$ $cm^{-3}$ at sea surface level, and we can estimate the dry deposition flux $F\_dry = - 4$ $cm^{-2} s^{-1}$ to $-80$ $cm^{-2}$ $s^{-1} = - 0.04$ to $- 0.8 \times 10^6$ $m^{-2} s^{-1}$.*

*On that day, the EC and K theory flux estimates close to the surface are within this dry deposition flux range, i.e $F_{EC,bottom} = -0.4 \times 10^6$ $m^{-2} s^{-1}$ and $F_{K,bottom} = - 0.05 \times 10^6$ $m^{-2} s^{-1}$. The surface flux estimated by MLG, $F_s = -18.8 \times 10^6$ $m^{-2} s^{-1}$, is about 25 times higher compared to the higher estimate. The entrainment flux $F_e = -0.3 \times 10^6$ $m^{-2} s^{-1}$ and the fluxes close to the top of the MBL are in the same order of magnitude.*

*We added to the manuscript:*

*'The estimated fluxes were furthermore compared with the dry deposition flux $F\_dry$ using the approach $F\_dry = - v\_dry * N$ (in $cm^{-2} s^{-1}$). From Emerson et al. (2020, PNAS), for 100 nm particles one can estimate a dry deposition velocity to water in the range of $v\_dry = 0.01$ cm/s to 0.2 cm/s. For flight #3 on July 5$^{th}$, the particle number concentration was about $N = 400$ $cm^{-3}$ at sea surface level leading to a dry deposition flux $F\_dry = - 4$ to $- 80$ $cm^{-2} s^{-1} = - 0.04$ to $- 0.8 \times 10^6$ $m^{-2} s^{-1}$. On that day, the EC and K theory flux estimates close to the surface are within this dry deposition flux range, i.e $F_{EC,bottom} = - 0.4 \times 10^6$ $m^{-2} s^{-1}$ and $F_{K,bottom} = - 0.05 \times$*

*$10^6$ m$^{-2}$ s$^{-1}$. The surface flux estimated by MLG, $F_s$ = -18.8 x $10^6$ m$^{-2}$ s$^{-1}$, is about 25 times higher compared to the higher estimate. The entrainment flux $F_e$ = - 0.3 x $10^6$ m$^{-2}$ s$^{-1}$ and the fluxes close to the top of the MBL are in the same order of magnitude.'*

*Emerson, E.W., Hodshire, A.L., DeBolt, H.M., Bilsback, K.R., Pierce, J.R., McMeeking, G.R., Farmer, D.K. (2020) Revisiting particle dry deposition and its role in radiative effect estimates. PNAS 117, 26076–26082. www.pnas.org/cgi/doi/10.1073/pnas.2014761117*

---

## Author Comment (AC3)

**Reply to Review 2 (RC3):**

The paper by Luckerath et al. addresses an important topic of aerosol dynamics in the boundary layer where large uncertainties exist in estimating particle fluxes either directly by flying platforms or by indirect/ground measurements. Although scientifically the study does not deliver substantial results, method comparison using specific experimental platform is very important in understanding advantages and limitations of different methods and their uncertainties. The paper is written and developed very well and should be suitable for publication after providing a better context and clarifying few details.

The study was performed over the Northeast Atlantic and the authors should be aware of the number papers over the same region which are relevant both methodologically as well as for their comparative value(Flanagan, Geever et al. 2005, Geever, O'Dowd et al. 2005, Ceburnis, O'Dowd et al. 2008, Ceburnis, Rinaldi et al. 2016)

*We thank the reviewer for the positive feedback and agreed that more marine studies could be mentioned here. They used different approaches and measured usually closer to the sea surface and are therefore focused on the lower marine boundary layer, but should be mentioned in the introduction.*

*RC1 requested results on sea spray emissions in Mace Head and we added the following sentence and refences: 'Results from sea spray emission studies are published in e.g., Geever et al. (2005); de Leeuw et al. (2011) and Ovadnevaite et al. (2014).'*

*Additionally, we added right after the last sentence: 'Particle number fluxes during nucleation events at the Irish coast were studied by Flanagan et al. (2005) while Ceburnis et al. (2016) investigated sources and sinks of aerosol particles at the same location.'*

Comments

Line 36 Very much disputed aspect that sea spray contributes little to aerosol number <300nm. Please refer to (Ovadnevaite, Manders et al. 2014, Xu, Ovadnevaite et al. 2021)

*The discussion the papers mentioned above is partly also focused on the chemical composition of sea spray, the role of organic mterial as well as its hygroscopicity. Since our study is focused on the number concentration, we modified the sentence regarding the sea spray aerosol production in the following way: 'While the larger accumulation mode (diameter > 300 nm) is dominated by sea spray aerosol, the contribution of sea spray to the particle diameter range smaller than 300 nm is evaluated differently in the literature (Zheng et al., 2018; Ovadnevaite et al., 2014; deLeeuwet al., 2011} and is therefore subject of further research.'*

Line 59. Azores are indeed a good location for flight-borne measurements, but is it perfect given dominant high-pressure systems, contributing to mixing? Methods confirm this by 2/3 of the campaign characterized by dry weather with low cloud fraction. Mid-latitude oceans on the other hand are dominated by low pressure systems.

*Agreed. We modified the statement in the revised version to: 'In previous studies it turned out that the islands of Azores provide a good location for studying the MBL with low anthropogenic influence.'*

Line 101. I wonder how much of the disturbance helicopter created during the ascent? Wouldn't the descent profile make more sense considering that external cargo was hanging below a helicopter?

*During ascent and descent, the helicopter has always a true airspeed of about 20 m/s. The downwash is therefore deflected backwards and the measurements are not influenced by the helicopter. See also the introductory paper by Siebert et al. 2006 where a sketch and more detailed arguments describe this issue.*

*We added the sentence:'During ascent and descent, the helicopter has always a true airspeed of about 20 m s- and the measurements are not influenced by the helicopter (Siebert et al. 2006).'*

Equation 3. Doesn't this formula produce unrealistic K values? E.g.
$K=0.3*0.3*20*48=86.4 m2/s$

*Thanks for this hint. We checked the values and realized that there was a mistake in the calculation of \tau, which was a factor of 10 to high. The time resolution of our data was 0.1 s, but obviously, they were used in the algorhythm like 1s-data, which create these unrealistic values.*

*Accordingly, we now use the correct value of \tau = 4.8 s. This modification leads to values for $F_K$, which are by a factor of 10 smaller. However, they are basically still within uncertainties of the other methods and the main conclusion does not change.*

*We modified accordingly: $F_K$ in all figures and in table 3 as well as the corresponding passages in the text.*

Line 164. Should vTAS be vair?

*Thanks for the hint! $v_{TAS}$ is the true air speed and is also the parameter used in Eq. 4. Therefore, we changed $v_{air}$ to $v_{TAS}$ in Eq. 4.*

Line 185. derivative instead of specification

*Thank you for this comment. We modified the sentence: "The Mixed Layer Gradient (MLG) method is also based on flux-gradient similarity, and derived from K-theory."*

Table 2. Comparison to environmental variables is lacking, like horizontal wind speed, etc.

*It is challenging to put more environmental variables for profiles into this table. The vertical distribution of e.g. wind direction and speed as well as temperature are described for the whole campaign in the overview paper by Siebert et al., 2021. Therefore, we refer to this paper for more details.*

*We added the sentence to the figure caption: 'An overview over the synoptic situation, meteorological parameters and their vertical profiles can be found in Siebert et al., 2021.'*

Ceburnis, D., C. D. O'Dowd, G. S. Jennings, M. C. Facchini, L. Emblico, S. Decesari, S. Fuzzi and J. Sakalys (2008). "Marine aerosol chemistry gradients: Elucidating primary and secondary processes and fluxes." Geophysical Research Letters **35**(7): L07804.

Ceburnis, D., M. Rinaldi, J. Ovadnevaite, G. Martucci, L. Giulianelli and C. D. O'Dowd (2016). "Marine submicron aerosol gradients, sources and sinks." Atmospheric Chemistry and Physics **16**(19): 12425-12439.

Flanagan, R. J., M. Geever and C. D. O'Dowd (2005). "Direct measurements of new-particle fluxes in the coastal environment." Environmental Chemistry **2**(4): 256-259.

Geever, M., C. D. O'Dowd, S. van Ekeren, R. Flanagan, E. D. Nilsson, G. de Leeuw and U. Rannik (2005). "Submicron sea spray fluxes." Geophysical Research Letters **32**(15): Artn L15810.

Ovadnevaite, J., A. Manders, G. de Leeuw, D. Ceburnis, C. Monahan, A. I. Partanen, H. Korhonen and C. D. O'Dowd (2014). "A sea spray aerosol flux parameterization encapsulating wave state." Atmospheric Chemistry and Physics **14**(4): 1837-1852.

Xu, W., J. Ovadnevaite, K. N. Fossum, C. S. Lin, R. J. Huang, C. O'Dowd and D. Ceburnis (2021). "Seasonal Trends of Aerosol Hygroscopicity and Mixing State in Clean Marine and Polluted Continental Air Masses Over the Northeast Atlantic." Journal of Geophysical Research-Atmospheres **126**(11).